# *UniFoil*: A Universal Dataset of Airfoils in Transitional and Turbulent Regimes for Subsonic and Transonic Flows

**Rohit Sunil Kanchi**[1]
Department of Mechanical and
Aerospace Engineering
University of Tennessee
Knoxville, TN 37996
rkanchi@vols.utk.edu

**Benjamin Melanson**[2]
Department of Mechanical and
Aerospace Engineering
University of Tennessee
Knoxville, TN 37996
bmelanso@vols.utk.edu

**Nithin Somasekharan**[3]
Department of Mechanical, Aerospace, and
Nuclear Engineering
Rensselaer Polytechnic Institute
110 8th St, Troy, 12180, New York, USA
somasn@rpi.edu

**Shaowu Pan**[4]
Department of Mechanical, Aerospace, and
Nuclear Engineering
Rensselaer Polytechnic Institute
110 8th St, Troy, 12180, New York, USA
pans2@rpi.edu

**Sicheng He**[5]
Department of Mechanical and
Aerospace Engineering
University of Tennessee
Knoxville, TN 37996
sicheng@utk.edu

## Abstract

We present *UniFoil*, the largest publicly available universal airfoil dataset based on Reynolds-averaged-Navier–Stokes (RANS) simulations. It contains over 500,000 samples spanning a wide range of Reynolds numbers, Mach numbers, capturing both transitional and fully turbulent flows across incompressible to compressible regimes. *UniFoil* is designed to support machine learning (ML) research in fluid dynamics, particularly for modeling complex aerodynamic phenomena. Most existing datasets are limited to incompressible, fully turbulent flows with smooth field characteristics, thus overlooking the critical physics of laminar–turbulent transition and shock-wave interactions-features that exhibit strong nonlinearity and sharp gradients. *UniFoil* fills this gap by offering a broad spectrum of realistic flow conditions. Turbulent simulations utilize the Spalart–Allmaras (SA) model, while transitional flows are modeled using an $e^N$-based transition prediction method coupled with the SA model. The dataset includes a comprehensive geometry set comprising over 4,800 natural laminar flow (NLF) airfoils and 30,000 fully turbulent (FT) airfoils, effectively covering the diversity of airfoil designs relevant to aerospace, wind energy, and marine applications. This dataset is also highly valuable for scientific machine learning (SciML), enabling the development of data-driven models that more accurately capture the transport processes associated with laminar–turbulent transition. *UniFoil* is freely available under a permissive CC-BY-SA license.

39th Conference on Neural Information Processing Systems (NeurIPS 2025) Track on Datasets and Benchmarks.

# 1  Introduction

Surrogate model-based aerodynamic shape optimization is a well-established field that has received significant attention over the past decades. Recently, machine learning (ML)-based surrogate models have shown substantial progress, offering the potential to accelerate design tasks. The objective is to develop an efficient and accurate model capable of rapidly identifying optimal designs. In particular, the surrogate must provide highly accurate drag predictions—within 1% relative error—across a broad range of flight conditions and geometric variations. However, existing efforts are often limited by small datasets with incomplete coverage of both the flight envelope and design space. To overcome these limitations, this paper introduces a comprehensive dataset comprising one million airfoil simulations spanning subsonic and transonic regimes, including both fully turbulent and laminar–turbulent transitional flows, a wide range of angles of attack (AoAs), and diverse geometric configurations.

In addition, another recent focus of the field is to discover new transport equation for laminar-turbulent transitional flow to replace the classic linear stability theory-based $e^n$ method. Our dataset also includes transitional airfoil simulations. This data can help to expedite the research in the data-driven transition model research.

The remainder of this paper is organized as follows. In Section 1, we review existing datasets and highlight the unique features of our proposed dataset, *UniFoil*. Section 2 details the tools and techniques used for data generation, while Section 3 describes the dataset structure and access details. Finally, conclusions are presented in Section 5. The dataset is available through the Harvard dataverse[1]. Tools to help view the dataset with preliminary ML results are available in the GitHub repository[2].

## 1.1  Related Work

Recent advances in ML for computational fluid dynamics (CFD) have led to the development of several open-access datasets tailored to automotive and aerospace applications. The AhmedML dataset Ashton et al. [2024b] offers hybrid RANS-LES simulations for 500 variants of the Ahmed body, supporting ML-based aerodynamic modeling. Similarly, WindsorML Ashton et al. [2024a] provides 355 wall-modeled large-eddy simulations (WMLESs) of the Windsor body to benchmark ML surrogate models. The DrivAerML dataset Ashton et al. [2024c] introduces high-fidelity HRLES simulations for 500 morphs of the DrivAer model. Meanwhile, AirfRANS Bonnet et al. [2022] focuses on 2D RANS simulations over airfoils, enabling ML exploration of steady-state incompressible flows. Finally, CFDGen Yagoubi et al. [2024] presents a synthetic data generation pipeline to enhance generalization and robustness of ML models across varying flow conditions. Collectively, these efforts underscore a growing commitment to reproducible ML research in CFD, though limitations remain in geometry complexity.

There are several airfoil datasets represent significant advancements in providing large-scale, ML-ready CFD data tailored for aerodynamic analysis. The datasets information is summarized in Table 1. The first dataset by Schillaci et al. [2021], comprises 425 simulations of turbulent flows over airfoils without modeling laminar–turbulent transition. The simulations are incompressible, and volume-resolved field data are provided for each case. The second dataset, AIRFRANS Bonnet et al. [2022], includes 1,000 simulations of turbulent airfoil flows at a Mach number (Ma) of 0.3 and Reynolds number (Re) ranging from 2 to 6 million. The dataset does not model transition and is limited to incompressible flows. It provides full field data across a wide range of angles of attack (AoAs). The third dataset by Ramos et al. [2023a], expands to 1,830 airfoils and incorporates both turbulence and transition modeling. Although the flows remain incompressible with a fixed Mach number of 0.1, the dataset spans multiple Re (3, 6, and 9 million) and AoAs from $-4°$ to $20°$ and includes over 250,000 simulations with corresponding flow field data. Other similar datasets have been summarized in Table 1 and compared with our *UniFoil* dataset.

---

[1]Harvard dataverse link - https://doi.org/10.7910/DVN/VQGWC4
[2]GitHub Repository — `https://github.com/rohitroxkp7/UniFoil`

Table 1: Comparison of publicly available 2D airfoil simulation datasets and our dataset.

| Dataset | No. of Airfoils | Turb. | Trans. Model | Compr. | Mach | Re (×10⁶) | AoA (°) | Field Data | No. of Sim. |
|---|---|---|---|---|---|---|---|---|---|
| Schillaci et al. [2021] | 425 | Yes | No | No | NA | 3 | 10 | Yes | 425 |
| Bonnet et al. [2022] | 1,000 | Yes | No | No | 0.3 | [2,6] | [-5,15] | Yes | 1,000 |
| Ramos et al. [2023a] | 1,830 | Yes | Yes | No | 0.1 | 3,6,9 | [-4,20] | Yes | 250,000 |
| Ramos et al. [2023b] | 9,000 | Yes | Yes | No | 0.1 | 9 | [4,12] | Yes | 17,992 |
| Agarwal et al. [2024] | 2,900 | Yes | No | No | NA | 0.1 | [-4,8] | No | 2,900 |
| Cornelius et al. [2025] | 18,000 | Yes | No | Yes | [0.25,0.9] | [0.075,8] | [-20,20] | Yes | 52,480 |
| **UniFoil (Current)** | **34,800** | **Yes** | **Yes** | **Yes** | **[0.1,0.85]** | **[1,10]** | **[-2,6]** | **Yes** | **500,000** |

## 1.2 Focus of this work

The existing dataset focuses on incompressible fully turbulent flows which missed critical strongly nonlinear physics including shock waves and laminar turbulent transition. While our new dataset is more general with these essential physics incorporated. More details can be found in Figure 1.

A shock wave is a narrow region across which flow properties such as pressure, temperature, and velocity change abruptly due to compressibility effects at high speeds (see the second figure in the first row of Figure 1). Shock waves introduce sharp spatial discontinuities in fluid flow that pose fundamental challenges for ML models, particularly those relying on smoothness assumptions or local interpolation. In high-speed regimes, shocks dominate the flow field and drive nonlinear phenomena such as boundary layer separation and wave interactions. For ML researchers, modeling and predicting shocks tests the limits of both inductive bias and network expressivity, especially in generalization across flow regimes. By including shock-resolving simulations in our dataset, we offer a benchmark that pushes beyond existing datasets constrained to subsonic, smooth flows—highlighting scenarios where naive ML architectures fail and physics-aware models are essential.

Laminar–turbulent transition presents a different but equally critical challenge: subtle, highly non-linear sensitivity to geometry, freestream conditions, and Re. Transition refers to the process in which a smooth, orderly (laminar) flow becomes chaotic and energy-dissipative (turbulent), often triggered by small disturbances that amplify through instability mechanisms. Unlike fully developed turbulence, transitional flows require ML models to detect weak, spatially diffuse signals that trigger instability growth. This makes the transition regime a compelling test case for evaluating data efficiency, uncertainty estimation, and transfer learning. By embedding detailed transition modeling in the dataset, we aim to foster research into hybrid physics-ML models capable of reasoning across regimes and learning representations that respect dynamic bifurcations—a capability that standard datasets with fully turbulent flows cannot test.

While high-fidelity CFD datasets with complex physical phenomena such as shock waves and laminar–turbulent transition are traditionally the focus of aerospace and applied physics communities, the growing intersection of scientific computing and ML necessitates their availability to the larger audience. The proposed dataset is designed not just for physical fidelity, but also for benchmarking modern ML methods—such as graph neural networks, physics-informed models, and latent-space surrogates—on realistic, multi-regime flow conditions. By introducing sharp nonlinearities (shocks) and regime-dependent sensitivities (transition), the dataset poses fundamental challenges in generalization, inductive bias, and physical consistency that are of direct interest to the NeurIPS community. In contrast to existing datasets that assume smooth flow fields, our dataset serves as a rigorous testbed for developing robust, generalizable models that can extrapolate across physics regimes, making it a valuable contribution to the ML for scientific discovery initiative.

## 1.3 Main contribution

In this paper, we aim to address several critical limitations of prior airfoil CFD datasets and contribute to advancing ML for high-fidelity fluid dynamics. The contributions of this work are summarized as follows:

- The largest publicly available high-fidelity CFD dataset of 2D airfoil flows, consisting of over **half-a-million simulations** spanning wide ranges of Re, Ma, and angle of attack (AoA);

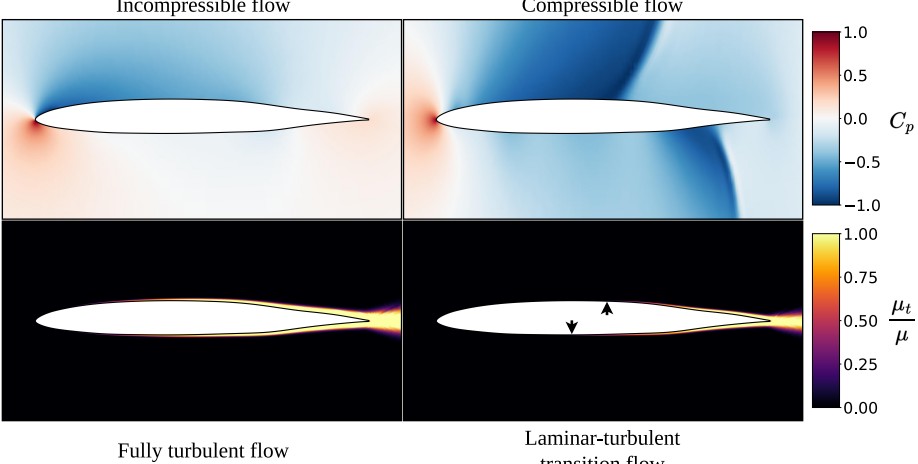

Figure 1: The first row shows the pressure coefficient ($C_p$) distributions for a subsonic flow at Mach 0.2 and a transonic flow at Mach 0.8. The subsonic case exhibits a smooth pressure field, whereas the transonic case features two shock waves forming on the suction and compression sides of the airfoil. The second row presents contours of the dimensionless turbulent viscosity, $\mu_t/\mu$, for both a fully turbulent model and a laminar–turbulent transition model under identical flight conditions. The fully turbulent model assumes immediate transition at the leading edge, while the transition model predicts a delayed onset of turbulence, with the transition location indicated by arrows.

- The first dataset of this scale to include both shock waves and laminar–turbulent transition phenomena, enabling ML models to be trained and evaluated on realistic, multi-regime aerodynamic behavior;
- Compressible RANS simulations using transition-sensitive turbulence models and shock-resolving numerical schemes;
- A comprehensive dataset including volume-resolved flow fields, wall-bounded quantities, integrated aerodynamic coefficients, and geometric descriptors in standardized formats;
- Full reproducibility enabled through open access to the simulation setup, meshing pipeline, and post-processing scripts, all built upon the extensively tested `ADflow` solver[3], which has been employed in multiple `NASA` projects.
- Released under a permissive CC-BY-SA license to support open science and integration into ML workflows.

To summarize and contrast the key new additions the current dataset brings to the literature, we refer the readers to Table 1 and a summary of the input fields can be found in Table 2.

## 2  Data Generation Methods

### 2.1  Geometry

Airfoil geometries can be efficiently represented in a reduced state space using modal decomposition techniques. In particular, a low-dimensional basis can be extracted from a dataset of classic airfoils via proper orthogonal decomposition (POD), enabling compact representation and efficient sampling of shape variations. Notably, airfoils designed for fully turbulent operation differ significantly in geometry—and consequently in modal characteristics—from those optimized to exploit laminar–turbulent transition mechanisms. Thus, we create two airfoil shape datasets for the fully turbulent (FT) airfoils and natural laminar flow (NLF) airfoils. In the current dataset, we sampled 30,000 FT and 4,800

---

[3]`ADflow` is a compressible flow solver developed by the University of Michigan's MDO Lab as part of the MACH-Aero framework. Source code is available at `https://github.com/mdolab/adflow`. For technical details, see Kenway et al. [2019].

NLF airfoils. An example of this sampling is shown in Figure 2 for the FT airfoils. We refer the readers to Li et al. [2019] for further information on sampling large datasets of geometric variations in a particular class of airfoils.

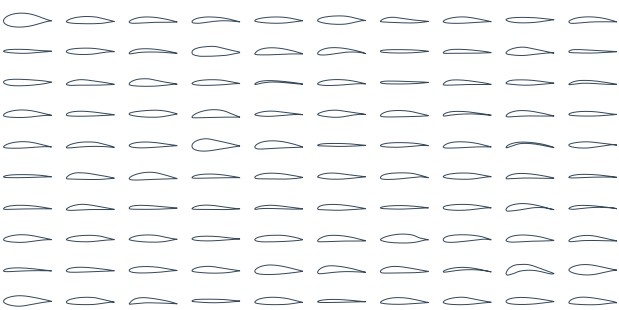

Figure 2: An overview of FT-airfoils geometry sampling.

## 2.2 Mesh generation

Mesh generation for the airfoil simulations was carried out using pyHyp by Secco et al. [2021][4]. pyHyp generates high-quality O-type multi-block meshes around airfoils using a hyperbolic marching method, which extrudes mesh lines from an initial surface mesh by solving a system of hyperbolic partial differential equations (PDEs) that control cell spacing, orthogonality, and smoothness. One mesh generated by pyHyp is shown in Figure 3. This approach enables accurate boundary layer resolution and consistent mesh quality, making it well suited for RANS solvers such as `ADflow`. The method is fully automated and robust across large geometric datasets, supporting high-throughput CFD simulations.

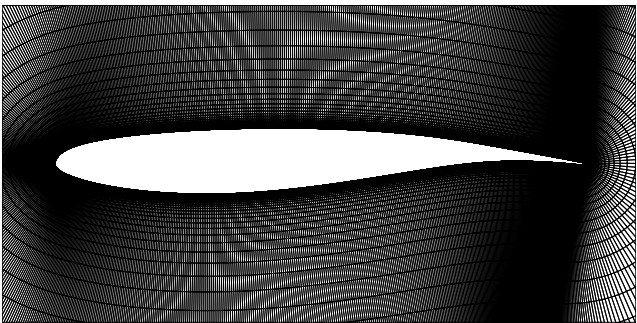

Figure 3: An example of the structured mesh around airfoil generated by pyHyp Secco et al. [2021].

## 2.3 Solver and numerical considerations

The `ADflow` solver was used in this study to simulate several thousand airfoils Mader et al. [2020]. `ADflow` is a structured, multi-block, compressible RANS solver that supports adjoint derivatives and is specifically designed for aerodynamic design and optimization. All turbulent flow simulations employed the Spalart–Allmaras (SA) turbulence model, while laminar–turbulent transition was modeled using an $e^N$-based transition prediction method coupled with the SA model, as recently implemented by Shi et al. [2020].

All simulations were steady state and performed on a far-field mesh with the airfoil body-fitted at the center of the mesh, as shown in Figure 3. The wall was given a no-slip boundary condition and the outermost edge of the computational domain was given a far-field boundary condition. Further details on the numerical methods employed in data generation are elaborated in the supplementary data.

---

[4]`https://github.com/mdolab/pyhyp`

An overview of the flow field figures and flow regimes in the dataset are summarized in Figure 1. In the figure, $C_p$ is the coefficient of pressure and $\mu_t/\mu$ is the ratio of the turbulent eddy viscosity to the fluid's molecular viscosity.

## 2.4  Convergence checks

The data quality was checked by monitoring the convergence of the numerical simulations and ensuring that all simulations in the dataset were sufficiently converged. It would be impractical to perform a mesh-dependency check on half-a-million simulations, and a fine mesh was therefore used in the data generation.

# 3  Dataset description

We sample a wide range of Mach numbers, Reynolds numbers, and AoAs. We have the largest ranges of Mach numbers and Reynolds numbers among all datasets Table 1. The range of AoA is narrower than in many other datasets, as large AoAs are typically avoided in practice to maintain aerodynamic efficiency. In addition, with a larger AoA, RANS model is no longer able to capture the flow field accurately due to flow separation. For each airfoil geometry, we conduct around 13 simulations with different Mach, Re, and AoA. These three input variables were sampled using a Lattice HyperCube Sampling approach.

The dataset is composed of three parts: The majority of the simulations are done using the fully turbulent flow assumption with SA model together with fully turbulent airfoil geometry class (more details see Section 2). This dataset represents the most commonly used operation condition. The second dataset also uses the fully turbulent assumption but uses the NLF geometry class. Finally, the last dataset has a one-to-one correspondance with the second dataset with identical flight conditions and geometries. The only difference is that the last dataset employs an $e^N$-based transition model to capture the laminar–turbulent transition, whereas the second dataset assumes fully turbulent flow from the leading edge of the airfoil.

Table 2: Statistics for the simulations in the *UniFoil* dataset.

| Turbulence Model | Airfoil Class | No. of Airfoils | Mach | Re ($\times 10^6$) | AoA ($^\circ$) | Net Simulations |
|---|---|---|---|---|---|---|
| SA model | FT | 30,000 | $[0.1, 0.85]$ | $[1, 10]$ | $[-2, 6]$ | 400,000 |
| | NLF | 4,800 | $[0.1, 0.85]$ | $[1, 10]$ | $[-2, 6]$ | 50,000 |
| $e^N$ transition model (+ trip term) | NLF | 4,800 | $[0.1, 0.85]$ | $[1, 10]$ | $[-2, 6]$ | 50,000 |

## 3.1  Dataset access and contents summary

The dataset is publicly available under the CC-BY-SA license at the Harvard Dataverse Kanchi et al. [2025]. The dataset contents are summarized in Table 3. All data processing and visualization scripts will be released in a public repository referenced in the supplementary material. The CGNS files listed in Table 3 have the coefficient of pressure, velocity vector data and the mach number. Each CGNS file retains the grid and the values of the computed flow variables in that grid. For the NLF-transition simulations, the CGNS files have additional information, such as turbulence intermittency, temperature, skin-friction coefficients and density. The lift and drag coefficients ($C_L$ and $C_D$) are placed in the tabular file as described in Table 3.

## 3.2  Computational resources

The simulations comprising this dataset were performed on the ANVIL cluster maintained by Purdue University. The computational setup included access to a total of 896 CPU cores. Each simulation was executed on a single core, enabling efficient parallelization across multiple batches. A dedicated 5 TB storage allocation was used for data management throughout the process. On average, each simulation required approximately 2 minutes to complete. The entire dataset was generated over a period of roughly 134,400 CPU Hrs.

Table 3: Overview of dataset structure and file contents.

| Component | Description |
|---|---|
| `Airfoil_Case_Data_turb.tab` | Mach, AoA, and Re for FT airfoil simulations. |
| `Turb_Cutout_<1-6>.zip` | FT airfoil simulations CGNS files. |
| `airfoil_dat_from_sim_turb<1-2>.tar.gz` | Convergence history and $C_L$, $C_D$ data for FT runs. |
| `Airfoil_Case_Data_Trans_Lam.tab` | Mach, AoA, and Re for NLF-turbulent and NLF-transitional simulations. |
| `NLF_Airfoils_Fully_Turbulent.zip` | NLF airfoils fully turbulent simulations CGNS files. |
| `airfoil_dat_from_sim_lam.tar.gz` | Convergence history and $C_L$, $C_D$ data for fully turbulent NLF runs. |
| `Transi_Cutout_<1-4>.zip` | Transitional simulations; each folder includes surface and volume CGNS files and slice .dat files. |
| `Transi_sup_data_cutout_<1-4>.zip` | Supplementary data from transitional simulations, including transition model outputs. |
| `airfoil_dat_from_sim_transi.tar.gz` | Convergence history and $C_L$, $C_D$ data for transition NLF runs. |
| `airfoil_geometry/` | Airfoil coordinates and generation code (in supplementary repo[2]). |

## 3.3 Limitations of the Dataset

While the *UniFoil* dataset provides a comprehensive and diverse set of airfoil simulations, it is important to acknowledge the following limitations:

- **Lack of time-accuracy:** All simulations in the dataset are steady-state and do not capture unsteady or time-accurate flow phenomena, which are often critical in real-world aerodynamic applications.

- **Two-dimensional approximation:** The simulations are performed in two dimensions, whereas real-life aerodynamic flows are inherently three-dimensional. This simplification may limit the applicability of the dataset for certain classes of problems involving spanwise effects or vortex shedding.

- **Absence of validation with high-fidelity or experimental data:** Although the simulations are conducted on sufficiently fine meshes (see supplementary material for details), the dataset has not been benchmarked against high-fidelity reference data such as direct numerical simulations (DNS) or wind tunnel experiments. This is primarily due to the computational cost associated with large-scale validation across the full dataset.

- **Unquantified geometric diversity:** While the airfoil shapes are derived from a POD of a large airfoil design space Li et al. [2019], a formal analysis comparing the geometric diversity of *UniFoil* against existing datasets has not been conducted. As a result, the extent to which the dataset spans the full range of practically relevant airfoil shapes remains to be quantified.

# 4    Preliminary results for ML

In this section, we present the key ML results using *UniFoil*. We describe the training architecture and present the predicted results bench–marked against results from `ADflow`. More details on data pre–processing and training and testing metrics can be found in Appendix E.

## 4.1    Neural network architecture

The neural network trained on the *UniFoil* dataset is an encoder-decoder model with a latent space deep neural network (DNN). An overview of the architecture of the neural network is shown in Figure 4. A summary of the training setup is shown in Table 4. In the table the "Param #" field represents the quantity of variables that can be modified during the training procedure. Densely connected layers such as the ones prevalent in the DNN have a large number of these parameters as all 6134 nodes are connected to every single node below. This model is trained on the pressure-coefficient data extracted from each airfoil simulation. The model is split into two main components with different layer designs used to down-sample and up-sample the input space. Further details on the exact architecture specifics and data pre–processing step are presented in Appendices E.1 and E.2.

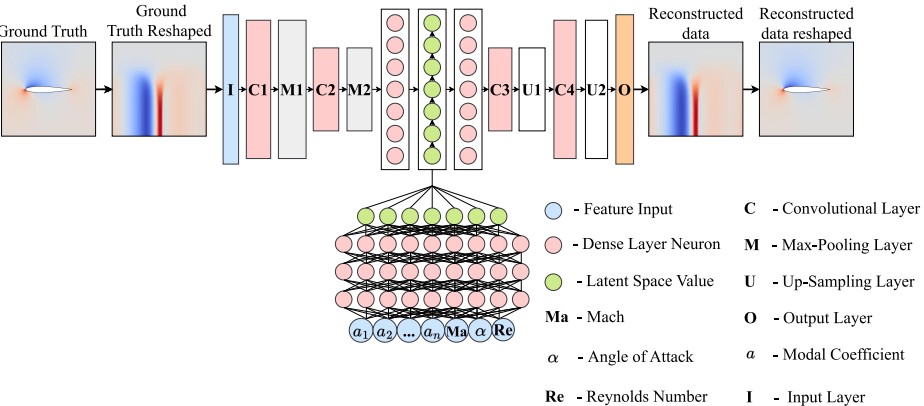

Figure 4: An overview of the entire training pipeline. The modal coefficients $a_i$ are obtained from Equation (5).

## 4.2    Training results

We present a visualization of the ground truth $C_p$ fields, reconstructed $C_p$ fields and the $L_2$ error for those plots for some of the airfoils randomly picked from the dataset in Figure 5. Appreciable comparisons can be seen for the field values between the ground truth and reconstructed $C_p$ data fields.

To further examine the model's performance, we plot the absolute error field (Figure 5),

$$\left| C_{p,\text{real}}^{(i)} - C_{p,\text{pred}}^{(i)} \right|, \tag{1}$$

which allows visualization of localized discrepancies. As shown in Figure 5, regions of high error are found in the vicinity of shock waves, particularly over the 3rd and 4th airfoils. These errors typically arise because the model struggles to accurately represent the steep gradients in pressure associated with shocks, resulting in localized deviations along the shock front.

Despite this limitation, the model reliably captures the overall structure of the pressure field, including the location of shocks and discontinuities. This makes the output useful for shock detection, even if exact magnitudes are not fully recovered. In contrast to the abrupt errors near shocks, more diffuse and persistent error patterns are evident in the background fields, such as around the 1st and 5th airfoils. These are attributed to a broader modeling challenge: the failure to accurately reconstruct mean pressure values far from the airfoil surface.

Table 4: Neural Network Architecture Summary

| Component | Layer Type | Output Shape | Param # |
|---|---|---|---|
| **Encoder** | Input | (32, 84, 292, 1) | 0 |
| | MaxPooling2D | (32, 42, 146, 1) | 0 |
| | Conv2D | (32, 42, 146, 32) | 320 |
| | MaxPooling2D | (32, 21, 73, 32) | 0 |
| | Conv2D | (32, 21, 73, 4) | 1,156 |
| | Flatten | (32, 6132) | 0 |
| | Dense | (32, 6132) | 37,607,556 |
| | Output | (32, 6132) | 0 |
| **Decoder** | Input | (32, 6132) | 0 |
| | Dense | (32, 6132) | 37,607,556 |
| | Reshape | (32, 21, 73, 4) | 0 |
| | Conv2DTranspose | (32, 21, 73, 32) | 1,184 |
| | UpSampling2D | (32, 42, 146, 32) | 0 |
| | Conv2DTranspose | (32, 42, 146, 8) | 2,312 |
| | UpSampling2D | (32, 84, 292, 8) | 0 |
| | Conv2DTranspose | (32, 84, 292, 1) | 73 |
| | Output | (32, 84, 292, 1) | 0 |
| **Deep Neural Network** | Input | (32, 17) | 0 |
| | Dense | (32, 6134) | 122,680 |
| | Dense | (32, 6134) | 37,632,090 |
| | Dense | (32, 6134) | 37,632,090 |
| | Dense | (32, 6134) | 37,632,090 |
| | Dense | (32, 6134) | 37,632,090 |
| | Dense | (32, 6134) | 37,632,090 |
| | Output | (32, 6134) | 0 |

One potential cause of this background error is the non-uniform resolution of the input data. The model receives unwrapped $C_p$ data from an O-grid representation, in which the regions closest to the airfoil surface are more densely sampled. As a result, the model becomes biased toward higher accuracy near the surface, while performance degrades with increasing distance.

Another contributing factor to elevated $L_2$ norm errors may be the small magnitude of the ground truth $C_p$ field in certain cases. Since the $L_2$ error is a relative measure, small denominator values in the error formula can amplify minor discrepancies in the numerator, leading to seemingly high error values.

To improve model performance and ensure consistency across varying geometries, two normalization strategies were applied. First, both the model outputs and the dataset were normalized to the range $[-1, 1]$. Second, we recorded the minimum and maximum $C_p$ values from each simulation and appended them to the model's latent space. This enabled the network to learn and predict the appropriate scaling factors directly from the geometry.

However, this approach introduces sensitivity: minor inaccuracies in the predicted extrema can propagate as large-scale errors across the entire field. This highlights a key limitation—without directly preserving the true range of the pressure coefficient values, the reconstructed fields may systematically deviate from the physical solution. Future work should explore alternative strategies to better preserve the physical fidelity of predicted $C_p$ distributions, particularly in the presence of shocks and weakly varying background fields. A detailed discussion, on the training and prediction steps, is in Appendices E.3 and E.4.

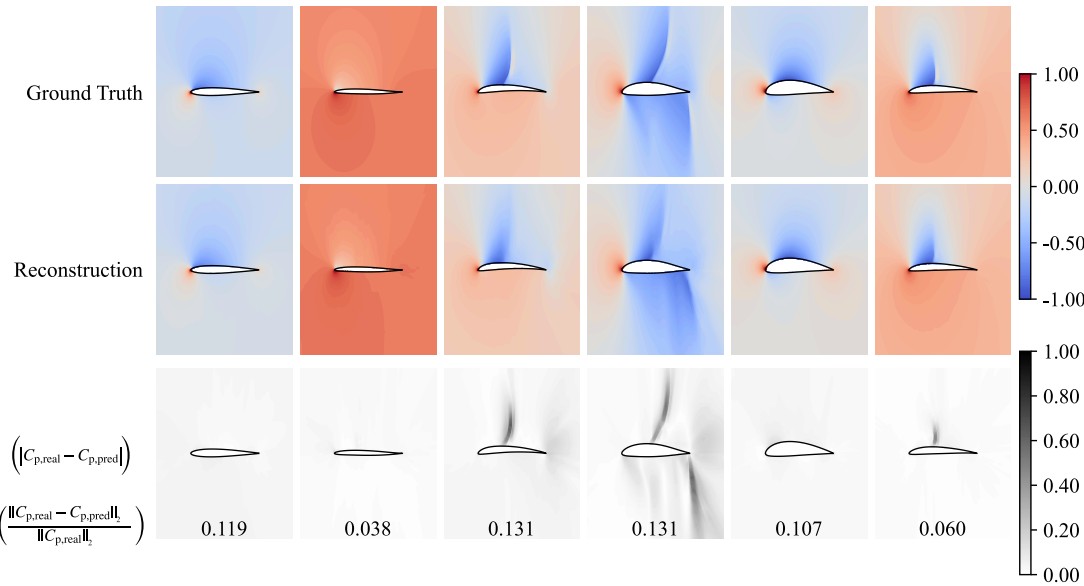

Figure 5: $C_p$ field contours from the ground truth data (first row) and reconstructed data (second row). Absolute error field is plotted in the third row. This is the absolute error between the pressure coefficient values from ADflow and prediction from neural network, as shown in the first formula from top in the third row. In the third row, the numbers in each sub–figure represent the relative $L_2$ norm error as defined in the second formula from top.

## 5  Conclusion

In this work, we introduce *UniFoil*, a comprehensive and large-scale dataset designed to bridge critical gaps in existing airfoil simulation datasets. By spanning a wide range of Reynolds numbers, Mach numbers, and AoAs, and by incorporating both laminar–turbulent transition and shock-wave phenomena, *UniFoil* provides a rich and realistic foundation for advancing ML in fluid dynamics. Unlike prior datasets that focus exclusively on incompressible and fully turbulent flows, *UniFoil* captures the full complexity of aerodynamic behavior across multiple physical regimes.

Through the use of compressible RANS simulations, transition-aware turbulence modeling, and shock-resolving numerical schemes, the dataset offers an unprecedented level of generality. Its standardized outputs—ranging from flow-resolved fields to integral coefficients–enable direct applicability across a wide spectrum of ML tasks, including surrogate modeling, uncertainty quantification, and regime classification.

By openly releasing the dataset, meshing pipeline, and post-processing scripts under a permissive CC-BY-SA license, we aim to foster open scientific collaboration and reproducibility. We anticipate that *UniFoil* will serve as a rigorous benchmark for testing the limits of current ML architectures and as a catalyst for developing physics-aware models that generalize across nonlinear, multi-regime aerodynamic phenomena.

## 6  Acknowledgement

This work used resources provided by the NSF grants PHY240112: Scalable Scientific Machine Learning for Complex Physical Systems and PHY240208: Data-driven Laminar–turbulent Transistion Model Discovery and Design Optimization, which provided the computational resources on Anvil at Purdue University for the dataset generation.

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

## A    Checklist

This section completes the checklist for the Neurips 2025 Datasets track submission.

- **Claims**:
  **Response** - **Yes**. The data presented in the manuscript in preceding sections suffices for the support of the paper's contributions and its scope.
- **Limitations**:
  **Response** - **Yes**. We have an explicit section (Section 3.3) in the paper that details the limitations of the dataset and re-inforces the scope of the manuscript.
- **Theory, Assumptions and Proofs**:
  **Response** - **NA**. The paper does not portray any theoretical results.
- **Experimental Result Reproducibility**:
  **Response** - **Yes**. The paper has a supplementary data section that provides details on how the data can be generated. The tools used in the dataset generation are all open-source.
- **Open Access to Data and Code**:
  **Response** - **Yes**. All data is open-access.
- **Experimental Setting/ Details**:
  **Response** - **Yes**. Details on the dataset structure and the input parameters to all simulations have been clearly provided in this manuscript in a file in the dataset.
- **Experiment Statistical Significance:**
  **Response** - **NA**.
- **Experiments Compute Resource**:
  **Response** - **Yes**. A section (Section 3.2) has been made to address this query.
- **Code Of Ethics**:
  **Response** - **Yes**.
- **Broader Impacts**:
  **Response** - **NA**.
- **Safegaurds**:
  **Response** - **NA**.
- **Licenses**:
  **Response** - **Yes**. All the work is published and available under the CC-BY-SA license.
- **Assets**:
  **Response** - **NA**.
- **Crowdsourcing and Research with Human Subjects**:
  **Response** - **NA**.
- **IRB Approvals**
  **Response** - **NA**
- **Declaration of LLM usage**
  **Response** - **NA**

# B  Geometry sampling

The modal representation is a useful tool to represent the geometry of the airfoil Li et al. [2019]. Taking a set of airfoil shapes, the modal analysis can extract the low dimensional modal representation that is efficient in capture the geometries. A single airfoil can be represented by $n$ surface points characterized by coordinates as $(x_i, y_i)$, $i = 1, 2, \ldots, n$. The $y$ coordinates for the $m$ airfoils can then be assembled into a matrix as

$$\mathbf{Y} = \begin{bmatrix} y_{11} & y_{21} & \cdots & y_{m1} \\ y_{12} & y_{22} & \cdots & y_{m1} \\ \vdots & \vdots & \ddots & \vdots \\ y_{1n} & y_{2n} & \cdots & y_{mn} \end{bmatrix}. \tag{2}$$

Performing singular value decomposition (SVD) in this matrix results in

$$\mathbf{Y} = \mathbf{U}\mathbf{\Sigma}\mathbf{V}^{\mathrm{T}}, \tag{3}$$

where $\mathbf{U} \in \mathbb{R}^{n \times n}$ contains the orthonormal spatial modes (airfoil shape modes) as columns, $\mathbf{\Sigma} \in \mathbb{R}^{n \times m}$ is a diagonal matrix of singular values, $\mathbf{V} \in \mathbb{R}^{m \times m}$ contains the parametric coefficients in its columns.

To reconstruct or generate a new airfoil shape using the leading $r$ modes ($r \ll n$), we define the reduced basis matrix as

$$\widetilde{\mathbf{U}} \in \mathbb{R}^{n \times r}, \tag{4}$$

where $\widetilde{\mathbf{U}}$ contains the first $r$ columns of $\mathbf{U}$. Then, any new airfoil shape $\mathbf{y} \in \mathbb{R}^n$ can be expressed as

$$\mathbf{y} = \widetilde{\mathbf{U}}\tilde{\mathbf{y}}, \tag{5}$$

where $\tilde{\mathbf{y}} \in \mathbb{R}^r$ is the vector of modal coefficients. It is noted that the $x$-values remain fixed and are termed "$x$-slices".

The choice of the modal-coefficients is key in generation of the airfoil shape database. Thus, instead of representing each airfoil with the $(x, y)$ coordinate values of its profile, we use the reduced few modal coefficients that were used to generate that airfoil in the input data to the neural network. For more details on choice of modal coefficients and how the sampling is done exactly for large number of airfoils, we direct the readers to Li et al. [2019, 2020]. In this manner, the $4,800$ NLF airfoils and $30,000$ FT airfoils were sampled. The code to sampling these airfoils and list of modal coefficients used for the generation of the dataset are detailed in the `GitHub` repository publicly available in the CC-BY-SA license[2].

# C  Numerical method

In this section, we discuss the numerical methods used in this paper. In the first subsection, we discuss the governing equations for the fully–turbulent and transition models used in the dataset generation for the FT and NLF airfoils. Then, we discuss the grid generation methods and input parameters used for the structured grids used in the simulations.

## C.1  Governing equations

We use `ADflow` as our CFD solver [Mader et al., 2020]. `ADflow` is a tool that has been extensively tested for a class of benchmark cases in the past. It solves the compressible Euler, laminar Navier–Stokes transitional and fully turbulent RANS equations using structured multi-block and overset meshes. `ADflow` has been used in aerodynamic, aero structural, and aero propulsive design optimization of aircraft configurations. Furthermore, we used `ADflow` to perform design optimization of hydrofoils and wind turbines.

The two equations that are considered in this research are the fully turbulent RANS equation with a SA turbulence model and the transitional RANS equation SA model coupled with the $e^n$ method [Shi et al., 2020]. The governing equation for compressible–RANS in conservation form is defined as follows

$$\frac{\partial \boldsymbol{q}}{\partial t} + \nabla \cdot \boldsymbol{f}_c(\boldsymbol{q}) = \nabla \cdot \boldsymbol{f}_v(\boldsymbol{q}, \nabla \boldsymbol{q}) + \boldsymbol{s}(\boldsymbol{q}), \tag{6}$$

where $\boldsymbol{f}_c(\boldsymbol{q})$ is the inviscid flux, $\boldsymbol{f}_v(\boldsymbol{q}, \nabla \boldsymbol{q})$ is the viscous flux, and $\boldsymbol{s}(\boldsymbol{q})$ is the turbulence source term, $\boldsymbol{q}$ contains the mean flow state variable and the turbulence variable, $\tilde{\nu}$ (turbulent SA eddy viscosity). The vector of conserved variables $\boldsymbol{q}$ includes the mean density, mean momentum components, mean total energy, and the working variable of the SA model in order as

$$\boldsymbol{q} = \begin{bmatrix} \bar{\rho} \\ \bar{\rho}\bar{u} \\ \bar{\rho}\bar{v} \\ \bar{\rho}\bar{E} \\ \tilde{\nu} \end{bmatrix}, \tag{7}$$

where $\bar{\square}$ denotes time–averaging, $\rho$ is density, $u, v$ are the $x$ and $y$ velocities, $E$ is energy and $\tilde{\nu}$ is the Spalart–Allmaras working variable used to compute the eddy viscosity as $\nu_t = \bar{\rho}\tilde{\nu}f_{v1}$, with $f_{v1} = \chi^3/(\chi^3 + c_{v1}^3)$, $\chi = \tilde{\nu}/\nu$ and $c_{v1}$ is an SA model constant.

The turbulence source term is defined as follows

$$\boldsymbol{s}(\boldsymbol{q}) = \begin{bmatrix} 0 \\ \boldsymbol{0} \\ 0 \\ g(C_{b1}(1 - f_{t2})\bar{S}\tilde{\nu} + \frac{C_{b2}}{\sigma}|\nabla\tilde{\nu}|^2 - C_{w1}f_w \left(\frac{\tilde{\nu}}{d}\right)^2) \end{bmatrix}, \tag{8}$$

where $C_{b1}, C_{b2}, \sigma, f_w, C_{w1}, d$ are SA model constants and $\bar{S}$ is defined as the magnitude of the mean vorticity tensor as

$$\bar{S} = \sqrt{2\,\Omega_{ij}\,\Omega_{ij}}, \qquad \Omega_{ij} = \frac{1}{2}\left(\frac{\partial \bar{u}_i}{\partial x_j} - \frac{\partial \bar{u}_j}{\partial x_i}\right) \tag{9}$$

where $S$ is the magnitude of the mean vorticity tensor, $\Omega_{ij}$ is the mean rotation rate tensor, and $\bar{u}_i$ is the mean velocity component in the $i$-th direction.

For more details, we refer the reader to the `NASA` website for more information ([NASA Langley Research Center]).

The transitional solver implements the $e^n$ transition model using the linear stability theory. An intermittency function $g \in [0, 1]$ is introduced to control the turbulence onset, which modified the source term of the turbulence equation as follows:

$$g\left(C_{b1}(1 - f_{t2})\bar{S}\tilde{\nu} + \frac{C_{b2}}{\sigma}|\nabla\tilde{\nu}|^2 - C_{w1}f_w \left(\frac{\tilde{\nu}}{d}\right)^2\right). \tag{10}$$

Here, if $g = 1$ the original SA equation is recovered and we are in the fully turbulent mode. If $g = 0$, the turbulence source term is completely turned off and we are in the fully laminar regime. When $g$ takes the value between 0 and 1, we are in the transitional region. The determination of $g$ is based on the $e^n$ method taking the information of the base flow.

Using the base flow information, once the transition location is found, the turbulence term will affect the main flow terms and thus resulting in a new equilibrium. An nonlinear block Gauss–Seidel method is used to handle the coupling between the RANS and the $e^n$ equations. More details on the transition modeling can be found in Shi et al. [2020].

## C.2   Grid generation

We use `pyHyp` Secco et al. [2021] to generate the volume grid, which uses a hyperbolic volume grid marching scheme to extrude structured surface grids into volume grids. Thus, the grid generation was performed such that the domain is an O-type grid with the airfoil body–fitted to the center of the domain with one cell thickness in the $z$-direction as shown in Figure 3.

The one-cell-thickness necessity arises from the fact that `ADflow` does not support pure two-dimensional simulations and therefore one needs to set one-cell thickness in the infinite span direction to assume two–dimensional (2D) nature of the flow. This is essentially a pseudo–2D simulation.

To maintain the same shape for the input data format, we ensured to maintain the same input parameters for the grid generation. The input parameters are summarized in Table 5. In the table, "TE" stands for trailing edge and "$c$" stands for chord, which was set to a value of 1m in all simulations.

Table 5: Fine grid generation parameters

| Description | Value |
|---|---|
| Number of points along airfoil surface till TE | 293 |
| Number of TE points | 11 |
| Number of points in marching direction | 85 |
| First layer thickness | $10^{-6}$ c |
| Distance from the airfoil surface to march the hyperbolic grid | $100c$ |

# D  Output file contents

As introduced in Section 3, the dataset comprises three categories of simulations:

1. Fully turbulent simulations for FT airfoils,
2. Fully turbulent simulations for NLF airfoils,
3. Transition simulations for NLF airfoils.

For each of the output files, we describe the contents in the following subsections. All simulation output files follow a consistent naming convention and are provided in CGNS format. The structure and content of these files are summarized in Table 6. In the following subsections, we discuss the output folders and files content into two sub–categories based on the geometry, the first being the FT–airfoil dataset and second being the NLF airfoil dataset.

Table 6: Overview of simulation data categories and file organization.

| Governing equation | Airfoil | Folder | Suffix | Freestream File |
|---|---|---|---|---|
| Fully Turbulent | FT | Turb_Cutout_<1-6>.zip | turb | Airfoil_Case_Data_turb.tab |
| Fully Turbulent | NLF | NLF_Airfoils_Fully_Turbulent.zip | lam | Airfoil_Case_Data_Trans_Lam.tab |
| Transition | NLF | NLF_Airfoils_Fully_Turbulent.zip | lam | Airfoil_Case_Data_Trans_Lam.tab |

## D.1  FT–airfoil dataset

The field data for FT airfoils is organized into multiple cutout folders named:

```
Turb_Cutout_<1-6>.zip.
```

Each simulation file follows the naming pattern:

```
airfoil_<airfoil_num>_G2_A_L0_case_<case_num>_000_surf_turb.cgns .
```

The integers `<airfoil_num>` and `<case_num>` refer to the specific airfoil geometry and the freestream condition, respectively. Detailed freestream conditions for each case are provided in the file:

```
Airfoil_Case_Data_turb.tab .
```

Each CGNS file contains the Mach number, pressure coefficient, $C_p$, and the velocity vector field. These files are compatible with open-source tools such as `ParaView` and `PyVista` library Sullivan and Kaszynski [2019] and commercial software such as `TecPlot` (commercial) for data extraction and visualization purposes.

A visualization of the field data present in the fully–turbulent dataset is shown in Figure 6. In the figure, we show the Mach number, velocity magnitude and pressure coefficient values. It also showcases shocks on both the suction and pressure sides, with flow separation aft the shock foot. The velocity magnitude $|\mathbf{u}|$ is computed using the formula

$$|\mathbf{u}| = \sqrt{u^2 + v^2}, \tag{11}$$

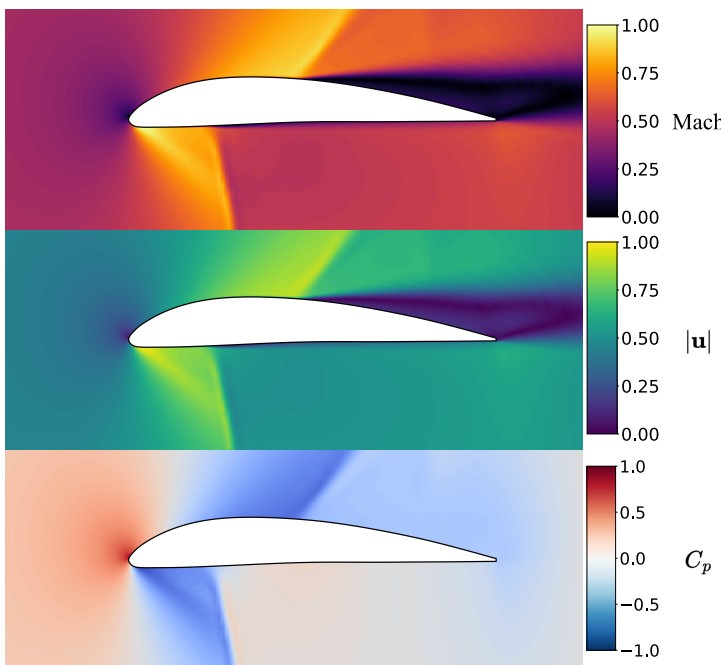

Figure 6: Plots of the data present in the fully turbulent simulations. $|\mathbf{u}|$ is the velocity magnitude and $C_P$ is the coefficient of pressure.

where $u$ is the $x$-velocity magnitude and $v$ is the $y$-velocity magnitude. Both these values including the velocity magnitude pre-computed through `ADflow` are available in the CGNS files.

### D.2 NLF–airfoil dataset

Two datasets are available for the NLF airfoils, the fully-turbulent dataset and the transition dataset. For both the datasets, the airfoils were run for the same freestream conditions.

The field data for NLF airfoils fully turbulent simulations is organized into a single folder named:



`NLF_Airfoils_Fully_Turbulent.zip` ,



with each simulation file inside this folder that follows the name:



`airfoil_<airfoil_num>_G2_A_L0_case_<case_num>_000_surf_lam.cgns` .



As aforementioned, the integers `<airfoil_num>` and `<case_num>` refer to the specific airfoil geometry and the freestream condition, respectively. Detailed freestream conditions for each case are provided in the file:



`Airfoil_Case_Data_Trans_lam.tab` ,



for both the fully-turbulent and transition datasets of the NLF–airfoils. Each CGNS file, once again, contains the Mach number, pressure coefficient, $C_p$, and the velocity vector field, as shown in Figure 6 for the FT-airfoils.

The transition simulation dataset however, follows a slightly different folder structure. Within each archive named `Transi_Cutout_<1-4>.zip`, the data is organized into subfolders following the format:



`airfoil_<airfoil_num>_G2_A_L0_case_<case_num>` .



Each of these subfolders contains the following files:

- CGNS files containing flow variables including the pressure coefficient, $C_p$, density, Mach number, and velocity field.
- A slice file in tabular format containing velocity and pressure field data extracted at the transition location along both the suction and pressure surfaces of the airfoil. These values represent a one-dimensional "slice" through the computational domain.

Finally, the archive `Transi_sup_data_cutout_<1-4>.zip` mirrors this folder structure and includes eight supplementary files per case. These provide extended data on the transition characteristics, enabling deeper analysis of the laminar-to-turbulent transition region. A full description of these supplementary files and their contents is available in the accompanying `README` file on the `GitHub` repository[2].

A visualization of the velocity magnitude field and the field of ratio of turbulent eddy viscosity to the molecular viscosity is shown in Figure 7. The transition onset points are marked on the airfoil surface using the red–colored arrow on the suction side of the airfoil. The transition to turbulence was only observed for the suction side in this case. The location for transition point was obtained from the "`transiLoc.dat`" file.

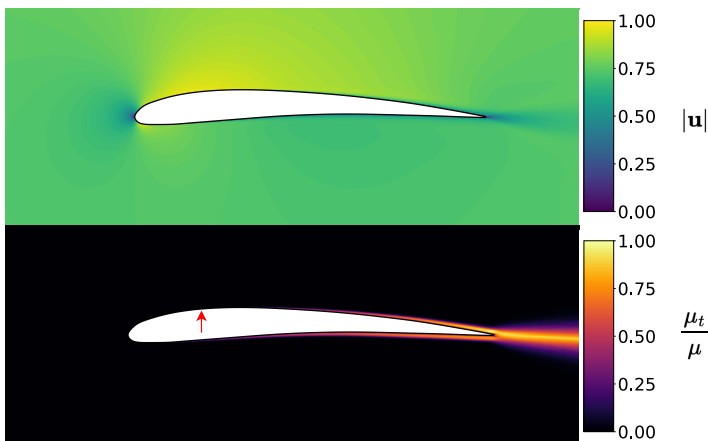

Figure 7: Velocity magnitude field and eddy viscosity ratio field in one of the transition flow simulations of the dataset. The laminar–turbulent transition point for the suction–side is located by the red arrow in the lower figure.

The $e^N$ method predicts the laminar separation and Tollmien–Schlichting (TS) waves induced transition mechanism. The TS waves amplification is characterized by a factor called $N_{\text{TS}}$ factor. Once it reaches a threshold of 7, the unstable wave is amplified in such a magnitude that the turbulence onset is triggered, which is when the flow transitions to fully–turbulent.

In the $e^N$ transition model, the reduced TS wave frequency is given by

$$f = \frac{\omega}{Re_\delta}, \tag{12}$$

where $\omega$ is the dimensionless angular frequency and $Re_\delta$ is the displacement–thickness based Reynolds number. If $\alpha_i$ is the local dimensionless spatial amplification rate for the TS wave, then the $N$ factor is given by

$$N = \int_{x_0}^{x} \alpha_i(x, f) \, \mathrm{d}x, \tag{13}$$

where $x$ is the spatial location in the computational domain and $x_0$ is the starting point of calculation. Thus, the $N_{\text{TS}}$ factor is defined as the maximum amplification among all waves

$$N_{\text{TS}} = \max_f(N(f)). \tag{14}$$

Out of the eight files, the key files are "`nfactor_ts.dat`" and "`transiLoc.dat`". The former has the TS wave factor $N_{\text{TS}}$ and the latter has the transition point location, if detected, on the suction and pressure surfaces. Details on accessing and visualizing these files can be found in the `GitHub`

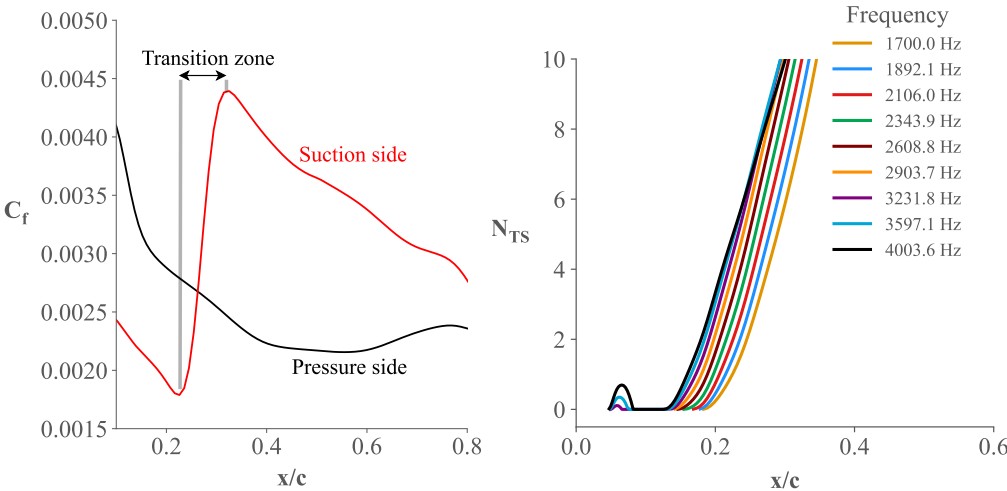

Figure 8: Skin-friction coefficient $C_f$ and the $N_{TS}$ factor vs $x/c$ where $c$ is the chord. The $N_{TS}$ factor is plotted for their different frequencies. The region before the transition zone is laminar flow region and region after the transition zone is the fully–turbulent flow region.

repository[2]. Details on the transition characteristics and the factors can be found in the paper by Shi et al. [2020].

Figures of the skin-friction coefficient $C_f$ and the $N_{TS}$ factor plotted against $x/c$ from the `nfactor_ts.dat` file is shown in Figure 8, for the airfoil shown in Figure 7. The $N_{TS}$ factor for TS waves is computed as shown in Equation (14). When transition occurs, the skin–friction coefficient starts to abruptly increase. The transition region highlighted in Figure 8 can be seen for the suction–side of the airfoil where the skin friction coefficient suddenly rises to a value of roughly 0.0043 before it starts to dip. For the $N_{TS}$ factor, values begin to increase around $x/c$ of 0.2, in–line with the observation from the skin–friction coefficient plot.

## E    Preliminary results for ML–Additional information

### E.1    Data pre–processing

To ensure a smooth handling of input data, maintaining the same input shape to the neural network and to preserve the grid resolution without loss of data that typically occurs when interpolating onto a uniform grid, the data from the O-grid was reshaped into a neat 2D array. The process of reshaping is highlighted in Figure 9. We start at the TE and move clockwise along the airfoil surface to loop till we reach back to the TE. This forms the first bottom–most layer of the 2D array. We then move to the next layer and repeat till all the layers are completed.

Since the same tool (`pyHyp`) and grid generation inputs from Table 5 were used, we maintained the same number of "O-grid rings" and ordering of points to reshape all input data in a similar manner. Figure 9 shows this concept in the top row and shows the implementation of the reshaping algorithm for the velocity field in the bottom row.

### E.2    Neural network architecture

The encoder consists of two smaller encoding blocks made of a convolutional layer and a max pooling layer. The decoder is the opposite and consists of two blocks made of a convolutional layer and a up-sampling layer. Once the model is trained, the encoder can be used to convert all of the data into the model's latent space. These latent space values are then saved and used to train the DNN. The train-test split used for this model was 80% / 20%. The model was trained for 100 epochs. Model batch size was 32. All neurons' activations were set to ReLU.

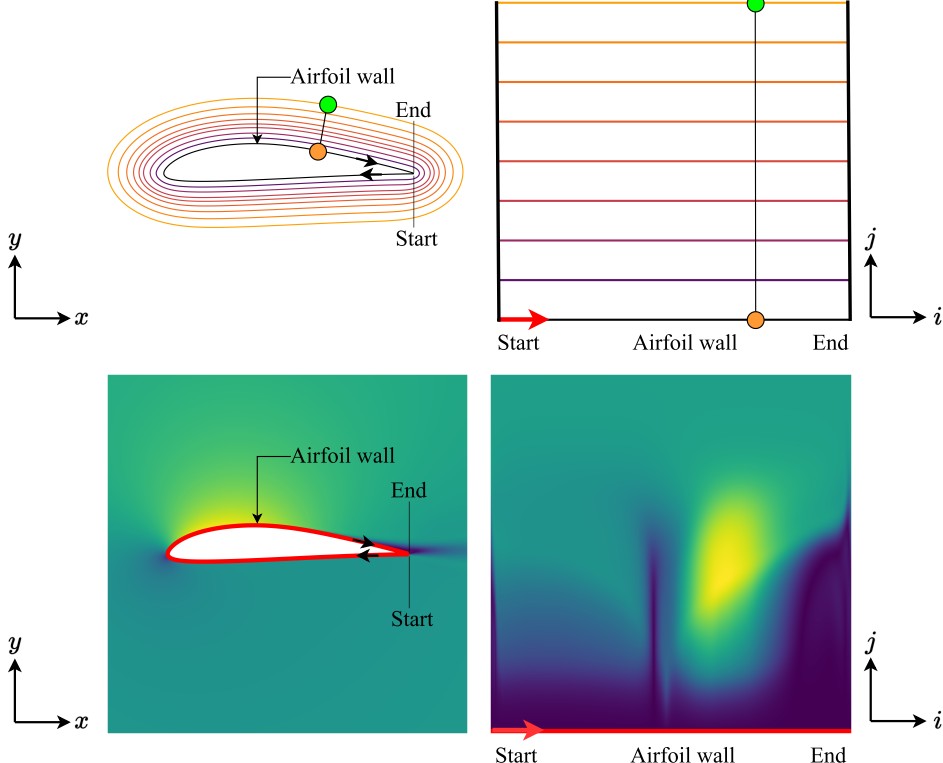

Figure 9: The input data is transformed from an O-grid mesh to a uniform rectangular grid by reordering the points in an array-like structure, without need to preserve their original spatial arrangement. The first row presents the concept and second row presents the application to the velocity–field data. The first column represents the data in the Cartesian coordinate system while the second column represents the data simply arranged in a 2D space using the node numbering, following the order of path guided by arrows in the figures.

The model was trained by first preparing the encoder-decoder model and training it on the reshaped pressure coefficient data. Once this model is trained, the encoder is used to convert all of the pressure coefficient data into latent space values which are saved separately. These latent space values are then combined with matching airfoil feature data in order to train the deep neural network. When the deep neural network is done training, the model can be used to predict an airfoils pressure coefficient field by first converting the feature data into the latent space and then passing this latent space data into the decoder. This results in a pressure coefficient field in the square grid format which must finally be reshaped into the O-Grid using the geometry data from the airfoil.

It is possible to simplify this method by cutting out the encoding step entirely and instead training a hybrid deep neural network and convolutional model. This can have a similar architecture but doesn't store the latent space information or a encoder allowing it to operate more efficiently. This approach should be tested in the future with different layer architectures to see if it produces more accurate results.

### E.3 Training

The neural network was trained using the FT-airfoils dataset. 400,000 simulations from the dataset were used for training. During the training process, the convolutional network and DNN errors were

plotted against the number of epochs. We use the following metrics for the loss function

$$
\text{loss}_{\text{encoder-decoder}} = \frac{1}{n_{\text{sample}}} \sum_i \frac{\left\| C_{p,\text{real}}^{(i)} - C_{p,\text{pred}}^{(i)} \right\|_2}{\left\| C_{p,\text{real}}^{(i)} \right\|_2},
$$
$$
\text{loss}_{\text{DNN}} = \frac{1}{n_{\text{sample}}} \sum_i \frac{\left\| C_{p,\text{real}}^{(i)} - C_{p,\text{pred}}^{(i)} \right\|_2}{\left\| C_{p,\text{real}}^{(i)} \right\|_2},
$$

(15)

where $C_{p,\text{real}}$ is the pressure coefficient from the ground–truth data and $C_{p,\text{pred}}$ is the pressure coefficient from the predicted data. A plot of the training error as shown in Equation (15) is shown in Figure 10. It is observed that the error steadily decreased for both networks and flattened out after 30 epochs for the DNN while still showing a slight but very slow decreasing trend for the convolutional network layer.

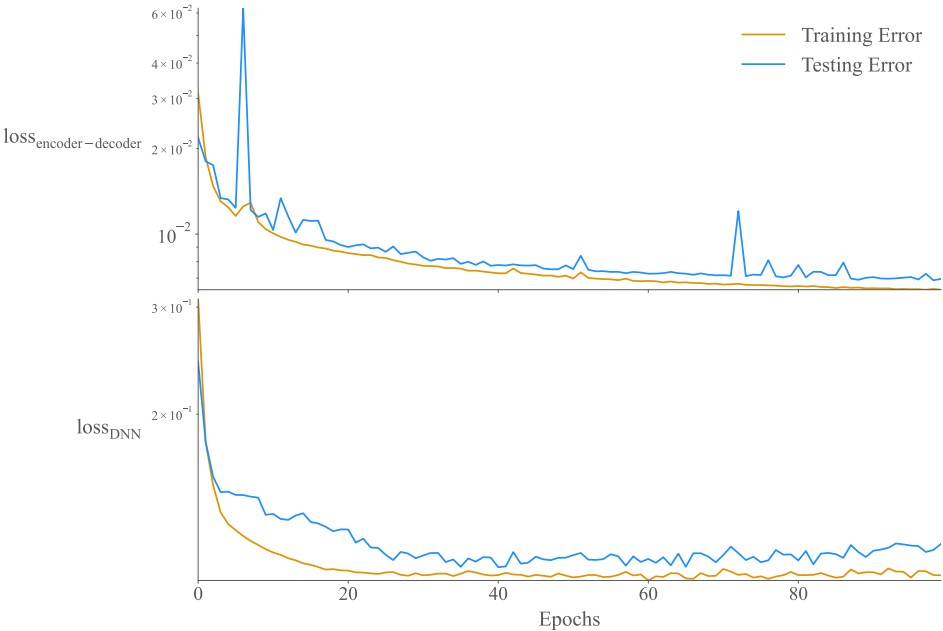

Figure 10: Training error against the number of epochs for the convolutional network and DNN.

## E.4  Prediction

Once the training was complete, a plot for the predicted values of pressure coefficient $C_{p,\text{pred}}$ against the ground truth values of $C_{p,\text{real}}$ was made for all the dataset simulations and their reconstructed field values.

This plot is shown in Figure 11 for a randomly picked airfoil. In an ideal case, the network would predict with no loss of accuracy. In such a scenario, we would have a one–one mapping of the predictions and ground truth data, yielding a straight line, $C_{p,\text{real}} = C_{p,\text{pred}}$. This is overlaid onto the plot as well for an ideal–case comparison. In the figure, it is observed that the prediction values were appreciably close to the ground truth values. In this case predicted $C_p$ values are greater then the ground truth near the high end of the plot while predicted $C_p$ values near the low end of the plot are less than the ground truth.

Next, to quantify the errors in the learning process, all of the input data was regenerated for the flow conditions they were simulated for. An $L_2$ norm error difference study between the actual data and the predicted data for each airfoil was then carried out and the number of airfoils quantity was plotted against the $L_2$ error for every airfoil in the 400,000 simulations and is shown in Figure 12. The

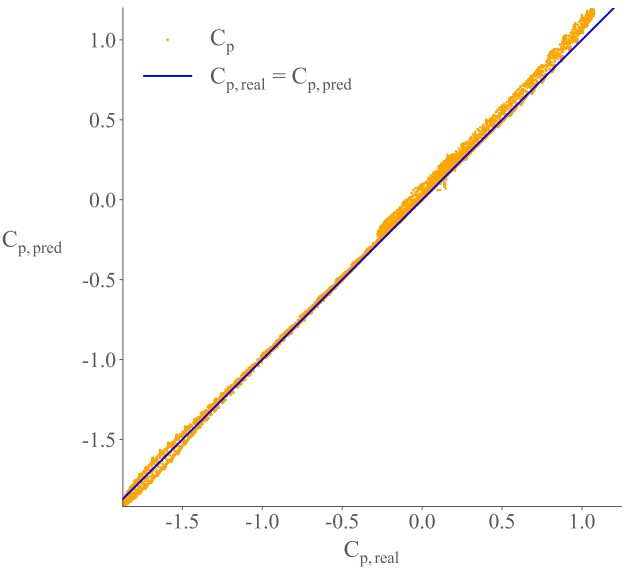

Figure 11: Predicted and ground truth values for $C_p$.

relative $L_2$ norm error is defined as follows

$$\frac{\left\|C_{p,\text{real}}^{(i)} - C_{p,\text{pred}}^{(i)}\right\|_2}{\left\|C_{p,\text{real}}^{(i)}\right\|_2}, \tag{16}$$

where $C_{p,\text{pred}}$ is the pressure coefficient values from the predicted data, the $C_{p,\text{real}}$ is the pressure coefficient from the ground-truth data and $i$ is the simulation count. This histogram plot gives an overview of the error distribution in the dataset. Roughly 35,000 airfoil simulations from the 400,000 had the highest error of approximately $10^{-1}$.

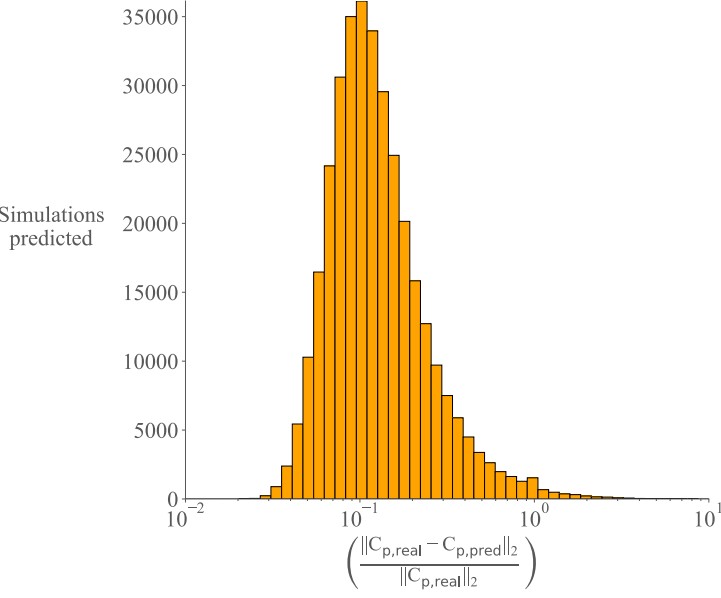

Figure 12: Number of airfoil simulations vs $L_2$ norm of the differences in $C_p$ between ground truth and predictions.