# OpenReview forum: "UniFoil: A Universal Dataset of Airfoils in Transitional and Turbulent Regimes for Subsonic and Transonic Flows"
_NeurIPS.cc/2025/Datasets_and_Benchmarks_Track — NeurIPS 2025 Datasets and Benchmarks Track poster_

### Official Review · Reviewer_ziXW · 2025-06-29

**Rating:** 4
**Confidence:** 3

**Summary:**

This paper introduces UniFoil, a large-scale, publicly available dataset comprising over 500,000 Reynolds-averaged-Navier-Stokes (RANS) simulations for 2D airfoils. The dataset is designed to provide comprehensive coverage of aerodynamic phenomena, including laminar-turbulent transition and shock-wave interactions, spanning a wide range of Reynolds numbers, Mach numbers, and angles of attack. It also incorporates a diverse set of 34,800 airfoil geometries.

While this work covers a comprehensive set of flow physics configs relevant to real-world applications, there is a significant lack of ML-based analysis and experiments to demonstrate the value of this dataset in an ML context. In its present form, I cannot recommend this paper for acceptance to an ML conference, unless there are ML applications and benchmarks performed here -- see previous year publications for examples for accepted papers.

**Dataset Code Accessibility:**

Partly

**Dataset Code Comments:**

While data links are provided, there aren't any code for loading/analyzing data provided -- making reproduciblity and widespread use cumbersome.

**Ethical Considerations:**

No, there are no or only very minor ethics concerns

**Final Justification:**

Raising to 4 since supp material addresses my concerns. However, ML portion is still insufficient for a stronger score

**Limitations Weaknesses:**

1. Lack of Direct ML Contribution: For an ML conference, the most significant weakness is the complete absence of any machine learning methodology, experiments, or baseline results using the dataset. The paper is purely a dataset description. It proposes to support ML research but does not demonstrate how this is achieved, nor does it identify specific ML challenges or provide benchmarks. This makes it difficult for ML researchers to immediately grasp the practical utility and specific research avenues the dataset enables from an ML perspective.

2. No ML Task Definition or Performance Metrics: The paper does not define any specific ML tasks that can be performed with UniFoil, nor does it provide baseline performance metrics for any ML model on these tasks. This omission is critical for an ML paper, as it leaves the reader to infer the dataset's direct utility for ML.

**Strengths Contributions:**

1. Comprehensive Coverage of Flow Conditions: The most significant strength is the sheer size (>500,000 simulations) combined with the inclusion of critical and complex flow physics often missing in other datasets, notably laminar-turbulent transition and shock waves. These phenomena introduce strong non-linearities and sharp gradients that are highly challenging for ML models and represent a crucial step towards more realistic aerodynamic modeling.

2. Relevance to Real-World Applications: By capturing transitional and compressible flows, the dataset addresses more realistic aerodynamic scenarios relevant to aerospace, wind energy, and marine applications, which can lead to more practical ML solutions.

---

> ### Author Rebuttal · Authors · 2025-07-31
>
> Rev - Lack of Direct ML Contribution: For an ML conference, the most significant weakness is the complete absence of any machine learning methodology, experiments, or baseline results using the dataset. The paper is purely a dataset description. It proposes to support ML research but does not demonstrate how this is achieved, nor does it identify specific ML challenges or provide benchmarks. This makes it difficult for ML researchers to immediately grasp the practical utility and specific research avenues the dataset enables from an ML perspective.
>
> No ML Task Definition or Performance Metrics: The paper does not define any specific ML tasks that can be performed with UniFoil, nor does it provide baseline performance metrics for any ML model on these tasks. This omission is critical for an ML paper, as it leaves the reader to infer the dataset's direct utility for ML.
>
> Response:
> We acknowledge the reviewer’s concern here. Several points in the comment have been addressed in the supplementary section of our manuscript. We apologize for the confusion caused by this placement of data. If permitted and necessary, we will move this data to the main section of the manuscript.
> In the supplementary data section, we have results for an encoder-decoder model with a latent space deep neural network which was trained on the UniFoil dataset. We detail (1) the ML methodology employed, (2) ML task definition, (3) baseline results, (4) performance metrics for the baseline results and (5) a short discussion, in the final section titled Prediction, on identification of specific ML challenges encountered and dealt with in the training/testing process. One key challenge was the non–linearity involved that arose from predicting the shockwaves which are sudden and sharp features in an otherwise smooth flow field.
> A key comment from the reviewer on Benchmark was not addressed. However, we use ADflow, a state of the art solver that has been used in aerodynamic, aerostructural, and aeropropulsive design optimization of aircraft configurations and has been verified by NASA projects [1,2]. The neural network in our study was benchmarked against the results from the solver used for the dataset. A challenge we currently face is the inconsistency in interface and data formats used across different ML methods. We aim to standardize this with UniFoil, to better aid in the benchmarking process across several ML platforms by creating a website.
>
> Rev - While data links are provided, there aren't any code for loading/analyzing data provided -- making reproduciblity and widespread use cumbersome.
>
> We appreciate the reviewer’s comment. However, we do have a GitHub repository titled ‘UniFoil’ and a link to this repository is available in the supplementary section of the paper as a footnote. To enable ease of quick repo access, we can move this to the main part of the manuscript if permitted. We have made this change in the arXiv version of our manuscript.
> The repository contains codes with sufficient documentation that contain (1) geometry sampling, grid generation for cfd, cfd workflow for data generation; (2) codes for loading and analysis of dataset files; (3) ML workflow scripts to enable end—users perform preliminary ML on the dataset to get started with. We created this repository for enabling easy use of UniFoil and reproducibility.
> We therefore encourage the reviewer to kindly review this repository, so we can further improve the dataset interface quality and enable ML research with UniFoil.
>
> References:
>
> [1] Mader, C. A., Kenway, G. K. W., Yildirim, A., & Martins, J. R. R. A. (2020). ADflow: An Open-Source Computational Fluid Dynamics Solver for Aerodynamic and Multidisciplinary Optimization. Journal of Aerospace Information Systems, 17(9), 508–527.
>
> [2] Yildirim, A., Jacobson, K. E., Anibal, J. L., Stanford, B. K., Gray, J. S., Mader, C. A., Martins, J. R. R. A., & Kennedy, G. J. (2025). MPhys: A modular multiphysics library for coupled simulation and adjoint derivative computation. Structural and Multidisciplinary Optimization, 68(1).

---

> > ### Comment · Reviewer_ziXW · 2025-08-02
> >
> > Acknowledged -- raising score from 2 to borderline accept 4 since some ML utility is shown. Please try to put these in the main paper instead of hiding it in appendix. Reviewers are not expected to look at supp material.
> >
> > However, analysis is till limited in comparison to some of the stronger D&B papers that demonstrate a wide range of models and metrics. This is what's missing from a stronger score

---

### Official Review · Reviewer_nzyK · 2025-07-03

**Rating:** 5
**Confidence:** 2

**Summary:**

The UniFoil dataset is a significant and valuable contribution to the field of scientific machine learning (SciML) for computational fluid dynamics (CFD). It aims to fill crucial gaps in existing datasets by providing a massive, diverse, and publicly available collection of 2D airfoil data, specifically addressing complex flow phenomena often overlooked.

**Dataset Code Accessibility:**

Yes

**Ethical Considerations:**

No, there are no or only very minor ethics concerns

**Final Justification:**

My main questions have been solved.

**Limitations Weaknesses:**

Reliance on RANS Approximations: While RANS is standard, it's an approximation of real fluid physics. The dataset's accuracy is inherently limited by the fidelity of the underlying RANS models used for its generation. This means it might not perfectly capture all intricate flow details that higher-fidelity simulations (like LES or DNS) could provide.

Computational Cost of Generation: Creating such a large and detailed dataset is extremely resource-intensive. While this isn't a weakness of the dataset itself, it highlights the immense effort required, and that similar datasets for even more complex scenarios would be similarly demanding.

Focus on 2D Airfoils: The dataset is specifically for 2D airfoils. While excellent for this domain, it doesn't extend to more complex 3D geometries (like full wings or aircraft components), which would require entirely different and even larger datasets.

Practical Accessibility for ML Workflows (Potential): While "publicly available," integrating very large scientific datasets into diverse ML frameworks can sometimes present practical challenges related to data parsing, storage, and efficient loading, though this would depend on the dataset's specific format and accompanying tools.

**Strengths Contributions:**

Fills Critical Gaps: Unlike many existing datasets, UniFoil includes crucial and complex flow phenomena like laminar-turbulent transition and shock-wave interactions. This makes it incredibly valuable for training ML models that need to handle realistic, non-linear aerodynamic behavior.

Broad Coverage: It spans a wide range of flow regimes (subsonic to transonic, incompressible to compressible), flow physics (transitional and fully turbulent), and airfoil geometries (thousands of natural laminar flow and fully turbulent airfoils). This diversity is key for developing generalizable models.

Massive Scale: With over 500,000 samples, it's a very large dataset, which is essential for training robust deep learning models that can capture complex physical relationships.

Realistic Data Generation: The data comes from RANS simulations using industry-standard models (Spalart-Allmaras and e^N transition method), making it highly relevant to engineering practice.

Open and Accessible: Being freely available under a permissive license greatly encourages its adoption and accelerates research across the community.

Direct Industrial Relevance: It specifically aims to enable ML models for highly accurate drag predictions across various flight conditions, directly addressing a core need in aerodynamic design and optimization.

---

> ### Author Rebuttal · Authors · 2025-07-31
>
> Rev - Reliance on RANS Approximations: While RANS is standard, it's an approximation of real fluid physics. The dataset's accuracy is inherently limited by the fidelity of the underlying RANS models used for its generation. This means it might not perfectly capture all intricate flow details that higher-fidelity simulations (like LES or DNS) could provide.
>
> Response:
> We thank the reviewer for this comment. It is appreciated that DNS and LES offer higher accuracy. However, their computational cost is prohibitive for generating such large-scale datasets. DNS becomes prohibitive beyond certain Mach and Reynold’s numbers and is expected to scale as Re^(2.91) [1].
> RANS has remained the gold standard in analysis of several fluid dynamical systems [2]. We also see verification and validation against experiments in top state of the art solvers such as SU2 [3] and ADflow [4], which is the one we used for UniFoil. Further, ADflow has been validated empirically for the range of flight conditions used in UniFoil in several NASA projects [5]. Thus, our goal here is rapid airfoil shape optimization through ML and several researchers have chosen the path of RANS for this task through the conventional gradient descent with solver approach [6,7]. Furthermore, among the many datasets cited in our manuscript that UniFoil is better than, AIRFRANS [8] is a dataset based on RANS published in NeurIPS.
>
> Our current agenda is to enable rapid airfoil shape optimization through ML driven models. Our ultimate goal is to construct a database of 3D unsteady high fidelity and large scale multiphysics data across several flow regimes to enable real time ML driven prediction of intricate flow physics across several standard external flows and internal flows such as flow over airfoil and flow in shaped pipes/ducts. UniFoil in its current state is the stepping stone for this vision.
>
> Rev - Computational Cost of Generation: Creating such a large and detailed dataset is extremely resource-intensive. While this isn't a weakness of the dataset itself, it highlights the immense effort required, and that similar datasets for even more complex scenarios would be similarly demanding.
>
> Response:
> We thank the reviewer for acknowledging our effort. Creating this dataset is indeed computationally expensive. However, the current landscape is fragmented—many research groups generate their own small, private datasets, which we believe is inefficient and limits broader progress. While our dataset requires significant effort to produce, we anticipate that it will ultimately reduce the overall effort across the ML community, for it has a wider range of flow regimes (incompressible-compressible and transition-fully turbulent) with a design to aid ML research.
> Furthermore, the use of ADflow, a solver extensively verified through multiple NASA projects [5], ensures both reliability and efficiency. Its structured grid formulation and advanced solver algorithms-including the Approximate Newton–Krylov (ANK) method-makes it one of the most effective tools available for generating large-scale aerodynamic databases [4].
>
> Rev - Practical Accessibility for ML Workflows (Potential): While "publicly available," integrating very large scientific datasets into diverse ML frameworks can sometimes present practical challenges related to data parsing, storage, and efficient loading, though this would depend on the dataset's specific format and accompanying tools.
>
> Response:
> We agree with the reviewer. A dataset of this size certainly presented challenges in terms of data handling and processing. To address these key challenges, we created a GitHub repository (link in supplementary section of manuscript) and have now included the end-end ML framework with extensive documentation to help ML users of UniFoil with a starting point.
> The codes address concerns of (1) data parsing, (2) storage (3) data loading and handling. The existing and tested framework in our repository uses Tensorflow and a .keras data storage format. The ML pipeline present in the repository with documentation for each step taken addresses all the concerns of the reviewer’s comment except one - while we do not have a (4) generalization to support different widely recognized ML frameworks, we plan to include this diversification for UniFoil.
>
> References:
>
> [1] Parviz Moin and Krishnan Mahesh. Direct Numerical Simulation: A Tool in Turbulence Research
>
> [2] Che Sidik, N. A., Yusuf, S. N. A., Asako, Y., Mohamed, S. B., & Aziz Japa, W. M. A. (2020). A Short Review on RANS Turbulence Models. CFD Letters, 12(11), 83–96
>
> [2] Zhoujie Lyu, Gaetan K.W. Kenway and Joaquim R. R. A. Martins. RANS-based Aerodynamic Shape Optimization Investigations of the Common Research Model Wing
>
> [3] Economon, T. D., Palacios, F., Copeland, S. R., Lukaczyk, T. W., & Alonso, J. J. (2016). SU2: An Open-Source Suite for Multiphysics Simulation and Design. AIAA Journal, 54(3), 828–846.
>
> [4] Mader, C. A., Kenway, G. K. W., Yildirim, A., & Martins, J. R. R. A. (2020). ADflow: An Open-Source Computational Fluid Dynamics Solver for Aerodynamic and Multidisciplinary Optimization. Journal of Aerospace Information Systems, 17(9), 508–527.
>
> [5] Yildirim, A., Jacobson, K. E., Anibal, J. L., Stanford, B. K., Gray, J. S., Mader, C. A., Martins, J. R. R. A., & Kennedy, G. J. (2025). MPhys: A modular multiphysics library for coupled simulation and adjoint derivative computation. Structural and Multidisciplinary Optimization, 68(1).
>
> [6] Li, J., Bouhlel, M. A., & Martins, J. R. R. A. (2019). Data-Based Approach for Fast Airfoil Analysis and Optimization. AIAA Journal, 57(2), 581–596.
>
> [7] Du, X., He, P., & Martins, J. R. R. A. (2021). Rapid airfoil design optimization via neural networks-based parameterization and surrogate modeling. Aerospace Science and Technology, 113, 106701.
>
> [8] Bonnet, F., Mazari, A. J., Cinnella, P., & Gallinari, P. (2022). AirfRANS: High Fidelity Computational Fluid Dynamics Dataset for Approximating Reynolds-Averaged Navier-Stokes Solutions. arXiv.

---

### Official Review · Reviewer_GisW · 2025-07-06

**Rating:** 4
**Confidence:** 3

**Summary:**

The paper present UniFoil, a publicly available dataset of 2D airfoil simulations based on compressible Reynolds-averaged Navier-Stokes (RANS) methods. The ADflow solver was used to for the flow simulation.
* The dataset contains half-a-million simulations.
* It contains a broad range of Reynolds numbers, Mach numbers, and angles of attack.
* It covers both transitional and fully turbulent flows across incompressible to compressible regimes.

**Additional Feedback:**

* The paper mentions "All simulations in the dataset are steady-state". If it's due to storage limitations for storing the data of each time step, would it be possible to provide code scripts that can generate the time-resolved data at each time step instead of including the full dataset directly?
* The github link (https://github.com/rohitroxkp7/UniFoil) is provided in the supplementary doc, it might be better to just put the github repo link in the main paper.

**Dataset Code Accessibility:**

Yes

**Dataset Code Comments:**

Dataset, code, and instructions to use the dataset/code are provided.

**Ethical Considerations:**

No, there are no or only very minor ethics concerns

**Final Justification:**

I decided to keep my original score 4 (Borderline accept), as the proposed benchmark dataset has a fair amount of contribution to the ML community.

**Limitations Weaknesses:**

* The dataset is somewhat limited in scope, as it only includes 2D cases and airfoil geometries.

**Strengths Contributions:**

* The authors provide details about the simulation setup, mesh generations, and post-processing steps.
* Compared to existing datasets that primarily focus on incompressible flows, UniFoil offers a larger volume of simulation samples and includes more phenomena such as shock waves and laminar–turbulent transition.
* The paper also demonstrates the effectiveness of the dataset in training ML models, by training and evaluating a baseline encoder-decoder models using the dataset.

---

> ### Author Rebuttal · Authors · 2025-07-31
>
> Rev - The dataset is somewhat limited in scope, as it only includes 2D cases and airfoil geometries.
>
> Response:
> We acknowledge the reviewer’s concern. However, our dataset focuses on airfoil shape, which by definition is 2D. Furthermore, enabling ML models to predict flow over an airfoil accurately would be a key success given current state of the art in Sci-ML [1]. Despite the simulation being 2D, we cover a wide range of flow regimes between incompressible and compressible, laminar-turbulent transition and fully turbulent flows; all of which do not exist in the current state of the art datasets, as shown in the Table 1 of our manuscript.
> Our current agenda is to enable rapid airfoil shape optimization through ML driven models. Our ultimate goal is to construct a database of 3D unsteady high fidelity and large scale multiphysics data across several flow regimes to enable real time ML driven prediction of intricate flow physics across several standard external flows and internal flows such as flow over airfoil and flow in shaped pipes/ducts. UniFoil in its current state is the stepping stone for this vision.
> ###
> Rev - The paper mentions "All simulations in the dataset are steady-state". If it's due to storage limitations for storing the data of each time step, would it be possible to provide code scripts that can generate the time-resolved data at each time step instead of including the full dataset directly?
>
> Response:
> We agree with the reviewer on this point. Unsteady simulations would significantly increase the computational cost in terms of storage space, effort and time. We use ADflow, a state of the art solver that has been used in aerodynamic, aerostructural, and aeropropulsive design optimization of aircraft configurations by several top research groups, including NASA [2,3]. Therefore, there are no appreciable limitations on the computational speed for the task at hand.
> However, the inclusion of URANS in the dataset requires careful consideration of physical time steps required for different airfoils and total physical time each simulation has to be run for. This puts a restriction on the number of airfoils that can be simulated in the same time frame UniFoil was generated. To give numbers, UniFoil in its current form took 134,400 CPU Hrs to be generated. If we were to run unsteady simulations in the given time frame, it would severely limit the design space and flight condition space to be sampled.
> Our current agenda is to enable rapid airfoil shape optimization through ML driven models and researchers have chosen the path of RANS for this task through the conventional gradient descent with solver approach [4,5] but our ultimate goal is to include unsteady simulations. UniFoil in its current state is the stepping stone for this vision.
> Code scripts for both steady and unsteady simulations using ADflow are now available in our GitHub repository.
> ###
> Rev - The github link (https://github.com/rohitroxkp7/UniFoil) is provided in the supplementary doc, it might be better to just put the github repo link in the main paper.
>
> Response:
> We thank the reviewer for the valuable feedback. We will add the link to the GitHub repository into the main part of the manuscript if permitted. We have made this change to our manuscript available on arXiv.
>
> References:
>
> [1] Brunton, S. L., Noack, B. R., & Koumoutsakos, P. (2020). Machine Learning for Fluid Mechanics. Annual Review of Fluid Mechanics, 52(1), 477–508.
>
> [2] Mader, C. A., Kenway, G. K. W., Yildirim, A., & Martins, J. R. R. A. (2020). ADflow: An Open-Source Computational Fluid Dynamics Solver for Aerodynamic and Multidisciplinary Optimization. Journal of Aerospace Information Systems, 17(9), 508–527.
>
> [3] Yildirim, A., Jacobson, K. E., Anibal, J. L., Stanford, B. K., Gray, J. S., Mader, C. A., Martins, J. R. R. A., & Kennedy, G. J. (2025). MPhys: A modular multiphysics library for coupled simulation and adjoint derivative computation. Structural and Multidisciplinary Optimization, 68(1).
>
> [4] Li, J., Bouhlel, M. A., & Martins, J. R. R. A. (2019). Data-Based Approach for Fast Airfoil Analysis and Optimization. AIAA Journal, 57(2), 581–596.
>
> [5] Du, X., He, P., & Martins, J. R. R. A. (2021). Rapid airfoil design optimization via neural networks-based parameterization and surrogate modeling. Aerospace Science and Technology, 113, 106701.

---

### Decision · Program_Chairs · 2025-09-18

**Decision:**

Accept (poster)

**Comment:**

UniFoil is a CFD dataset, useful for 2D training of neural operators